# Unraveling Signatures of Local Adaptation among Indigenous Groups from Mexico

**DOI:** 10.3390/genes13122251

**Published:** 2022-11-30

**Authors:** Humberto García-Ortiz, Francisco Barajas-Olmos, Cecilia Contreras-Cubas, Austin W. Reynolds, Marlen Flores-Huacuja, Meradeth Snow, Jazmín Ramos-Madrigal, Elvia Mendoza-Caamal, Paulina Baca, Tomás A. López-Escobar, Deborah A. Bolnick, Silvia Esperanza Flores-Martínez, Rocio Ortiz-Lopez, Aleksandar David Kostic, José Rafael Villafan-Bernal, Carlos Galaviz-Hernández, Federico Centeno-Cruz, Alejandra Guadalupe García-Zapién, Tulia Monge-Cázares, Blanca Patricia Lazalde-Ramos, Francisco Loeza-Becerra, María del Carmen Abrahantes-Pérez, Héctor Rangel-Villalobos, Martha Sosa-Macías, Augusto Rojas-Martínez, Angélica Martínez-Hernández, Lorena Orozco

**Affiliations:** 1Instituto Nacional de Medicina Genómica, Tlalpan, Mexico City 14610, Mexico; 2Department of Anthropology, Baylor University, Waco, TX 76706, USA; 3Department of Anthropology, University of Montana, Missoula, MT 59812, USA; 4Section for Evolutionary Genomics, The GLOBE Institute, The University of Copenhagen, Øster Farimagsgade 5A, 1352 Copenhagen, Denmark; 5Department of Anthropology and Institute for Systems Genomics, University of Connecticut, Storrs, CT 06269-3003, USA; 6División de Medicina Molecular, Centro de Investigación Biomédica de Occidente, Instituto Mexicano del Seguro Social (IMSS), Guadalajara 44340, Mexico; 7Tecnologico de Monterrey, Escuela de Medicina y Ciencias de la Salud and Insitute for Obesity Research, Monterrey 64700, Mexico; 8Centro de Investigacion y Desarrollo en Ciencias de la Salud, Universidad Autonoma de Nuevo Leon, Monterrey 64460, Mexico; 9Joslin Diabetes Center, Harvard Medical School, Boston, MA 02115, USA; 10Instituto Politécnico Nacional, CIIDIR-Durango, Durango 34220, Mexico; 11Departamento de Farmacobiología, Centro Universitario de Ciencias Exactas e Ingenierías, Universidad de Guadalajara, Guadalajara 44430, Mexico; 12Unidad Académica de Ciencias Químicas, Universidad Autónoma de Zacatecas, Zacatecas 98160, Mexico; 13Universidad Michoacana de San Nicolás de Hidalgo, Morelia 58030, Mexico; 14Instituto de Investigación en Genética Molecular, Universidad de Guadalajara Ocotlán, Ocotlán 44100, Mexico

**Keywords:** Native American populations, local adaptation, *PPARG*, *AJAP1*, gut microbiome

## Abstract

Few studies have addressed how selective pressures have shaped the genetic structure of the current Native American populations, and they have mostly limited their inferences to admixed Latin American populations. Here, we searched for local adaptation signals, based on integrated haplotype scores and population branch statistics, in 325 Mexican Indigenous individuals with at least 99% Native American ancestry from five previously defined geographical regions. Although each region exhibited its own local adaptation profile, only *PPARG* and *AJAP1*, both negative regulators of the Wnt/β catenin signaling pathway, showed significant adaptation signals in all the tested regions. Several signals were found, mainly in the genes related to the metabolic processes and immune response. A pathway enrichment analysis revealed the overrepresentation of selected genes related to several biological phenotypes/conditions, such as the immune response and metabolic pathways, in agreement with previous studies, suggesting that immunological and metabolic pressures are major drivers of human adaptation. Genes related to the gut microbiome measurements were overrepresented in all the regions, highlighting the importance of studying how humans have coevolved with the microbial communities that colonize them. Our results provide a further explanation of the human evolutionary history in response to environmental pressures in this region.

## 1. Introduction

Recent studies have shown that the initial settlement of America was characterized by a wide and rapid expansion throughout the continent by the first inhabitants who had, in many cases, to adapt to new and extreme environments [1,2,3]. This constant adaptation to diverse ecosystems, as well as complex demographic events, such as migrations and cultural and lifestyle transitions, contributed to the selection of alleles and to the diversity observed in Native American populations [4,5,6,7,8]. Nevertheless, only a few studies have explored how selective pressures have shaped the genetic structure of present-day Native American populations [9,10,11,12]. Such studies have identified strong selective signals at different genes, particularly at those related to metabolism and immunity. These highlight the selective pressures that different Native American communities were subjected to as a result of ancestral starvation periods and nomadic lifestyles as well as the challenge of infectious diseases brought by Europeans [6,7,9,11,12]. Several authors have stated that certain diseases, such as metabolic and immune entities, seem to be mediated by alleles that were targeted by natural selection in the past but have become detrimental in present-day populations due to recent and substantial cultural and lifestyle shifts [13,14]. Nevertheless, most studies of selection in the Americas have limited their inferences to the ancestry-masked genomes of Latin American admixed populations to investigate the loci under selection that are derived from their Native American ancestry [9,11,12].

To understand how selective pressures have shaped the genetic structure of Native American populations without the potential bias of recent admixture, we searched for local adaptation signals in present-day Mexican Indigenous individuals with the highest Native American ancestry studied to date (≥99%). A strong selective pressure related to metabolic and immune processes was found in all the groups, but evidence of a regional adaptation was also observed. Notably, the genes related to the gut microbiome measurement were also found to be under selection.

## 2. Materials and Methods

### 2.1. Samples

We included 325 individuals from the Metabolic Analysis in an Indigenous Sample (MAIS) cohort previously genotyped with the Affymetrix Human 6.0 array. Individuals having at least 99% Native American ancestry previously inferred by ADMIXTURE analysis, were selected from the MAIS Cohort [8]. Furthermore, a Neighbor-Joining Tree based on pairwise fixation index (*F_ST_*) values showed that those Indigenous groups from Mexico can be clustered in five regions: North, Northwest, Center, South, and Southeast [8].

### 2.2. Genome-Wide Scan for Selection Signals

A total of 654,023 SNVs were included for genetic analyses. To detect signatures of selection in these individuals, we followed the approach reported by Reynolds et al. [12]. We computed two statistical algorithms: the integrated haplotype score (iHS), a measure of extended homozygosity in the haplotype surrounding a given single-nucleotide variant (SNV), and the population branch statistics (PBS), which measures the amount of genetic differentiation at a given locus along a branch leading to a population of interest by comparing transformed *F_ST_* values between each pair of three populations (target, ingroup, and outgroup).

For iHS, the genotype data from the 325 individuals were phased using SHAPEIT4 v4.2.2 [15], with default parameters and the 1000 Genomes Project (1KGP) phase 3 dataset as a reference panel [16]. These phased genomes were annotated with ancestral allele information using aa_annotate.py [17]. After this, iHS values were calculated for each of the five geographic regions using hapbin v1.3.0 [18].

PBS calculation was performed following the methods proposed by Yi et al. [19]. Accordingly, *F_ST_* values were calculated for all five geographic groups with an ingroup composed of the 33 Peruvians with the highest Native American ancestry [12] and an outgroup composed of 50 Han Chinese (all of them from 1KGP phase 3) using vcftools v0.1.17 [20]. Finally, PBS was calculated for the five geographical regions.

Variants within the top 1% cross-referenced values for both PBS and iHS were considered as candidate genome locations for natural selection in each tested region.

### 2.3. Enrichment Analysis

Variants considered under selection in each region were annotated by the Ensembl Variant Effect Predictor [21]. With the list of genes harboring those selected variants, we generated the protein–protein interaction (PPI) network by region using the Search Tool for the Retrieval of Interacting Genes/Proteins database (STRING v.11.5, https://string-db.org; accessed on 15 August 2022) [22]. The parameters for this analysis were *Homo sapiens*, medium confidence, and hidden disconnected nodes in the network. With these networks, we analyzed the overrepresentation of phenotypes with the module Human Phenotype Ontology (MONARCH). Otherwise, to identify the potential overrepresentation of selected genes in any disease, we performed this analysis using the DAVID Bioinformatics Resources v2022q2 (david.ncifcrf.gov/; accessed on 15th August 2022) [23], with the parameters hsapiens, functional annotation tool, disease, and GAD_DISEASE. In both analyses, only terms with false discovery rate (FDR) < 0.05 were examined.

## 3. Results

### 3.1. Samples Description

The 325 individuals with 99% Native American ancestry belong to 53 Indigenous communities. The individuals were divided into the five previously defined regions [8] as follows: 51 from North, 26 from Northwest, 83 from Center, 71 from South, and 94 from Southeast (Figure 1; Appendix A).

### 3.2. Identification of Adaptation Signals

Overall, we found different candidate genes under selection in each geographic region, ranging from 78 in the South to 163 in the North (Figure 2; Appendix A). Several are related to the metabolic processes and immune response (Appendix A), although each region exhibited its own local adaptation profile (Figure 2A). In general, we observed that the populations from adjacent regions share more local adaptation signals than those from distant regions (Figure 2B, Appendix A), in line with previous observations regarding the genetic relationships between Indigenous populations from different geographic locations [8,9,12]. For example, the South shared 57 candidate genes with the Southeast but only 7 with the North (Figure 2C, Appendix A). Interestingly, only two genes showed significant adaptation signals in all the tested regions. These genes were peroxisome proliferator-activated receptor γ (*PPARG*) and adherens junctions–associated protein 1 (*AJAP1*) (Figure 2B, Appendix A).

### 3.3. Protein–Protein Interaction (PPI) Network of Genes under Selection

Next, we used STRING v.11.5 to construct the networks from the genes under selection in each region. In all the regions, a significant PPI enrichment was found (Figure 3 and Appendix A), documenting more interactions than expected for a random set of proteins. Several proteins were notable for their high degree of connection in every region. These included PIK3CG, PIK3C3, and PLCB1 in the North (Figure 3A and Appendix A); PPARG and CASP1 in the Northwest (Figure 3B and Appendix A); GRIN2B, PIK3C3, and GRIA1 in the Center (Figure 3C and Appendix A); CNTNAP2, FBN2, NCAM1, and LRRK2 in the South (Figure 3D and Appendix A); and PTPRD, GRIA1, and CADM2 in the Southeast (Figure 3E and Appendix A).

Regarding *PPARG* and *AJAP1*, the only genes with significant adaptation signals in all the tested regions, our analysis revealed PPARG to be located in the main networks constructed for the North, Northwest, and Center (Figure 3A–C and Appendix A) and to interact with the main network in the South and Southeast (Figure 3D,E and Appendix A). AJAP1 forms part of the main network from the Center (Figure 3C and Appendix A), and it interacts closely with the main network in the remaining regions (Figure 3 and Appendix A).

### 3.4. Enrichment Pathway Analysis of the Network of Genes under Selection

We also used the DAVID Bioinformatics Resources v2022q2 (david.ncifcrf.gov/; accessed on 15th August 2022) to identify a potential overrepresentation of the selected genes related to several biological phenotypes/conditions. Among them, the immune response, metabolic pathways, cardiovascular phenotypes, central nervous system, chemical dependence, response to stimulus, pulmonary function, hematological conditions, body weight, and brain and gut microbiome measurements were the common terms found to be enriched in the putatively selected genes in all the regions (Appendix A). However, the genes enriching the pathways were different from those reported for other populations, such as European or Japanese populations [24,25,26,27]. Some are exclusive to one or more geographic region, revealing the putative signatures of the local adaptation in Mexican Indigenous populations (Figure 2C; Appendix A).

One of the overrepresented biological pathways observed in all the regions was the gut microbiome measurements (Appendix A). Although the number of genes associated with the microbiome features, including the operational taxonomic unit (OTU) abundance, prevalence, or alpha/beta diversity, varied in each region, some genes were shared among the regions (Appendix A). These include *SLC6A11*, which has been associated with the *Oscillospira* and *Barnesiella* genera abundance [28,29] and was detected in the Center, South, and Southeast populations. Another is *TMTC2*, which affects the abundance of the genus *Faecalibacterium* [30] and was observed only in the North and Northwest (Appendix A). These data reflect the possible differences in the gut microbiome composition according to the geographic location and also the genetic profile of each region.

## 4. Discussion

The history of humankind is marked by the constant migration to new environments that provide new adaptive pressures for human physiology and may even contribute to the development of some human diseases [14,24]. The exposure to extremely diverse environments and subsequent environmental changes resulting from cultural innovations, such as the expansion of agriculture, have given rise to further selective pressures related to pathogen exposure and dietary shifts.

Geographic, demographic, and cultural phenomena that have led to peculiar patterns and heterogeneous genomic backgrounds in human population structures have been described previously [4,5,24,27,31]. A fine dissection, through the identification of genetically homogeneous clusters, may reveal functional genomic regions resulting from selective pressures that have favored the adaptation of populations to diverse environments during human natural history.

Based on these observations, we performed a genome-wide scan to detect the signatures of selection in the genomes of Mexican Indigenous individuals from five geographically defined groups from across Mexico (North, Northwest, Center, South, and Southeast), as previously described [8]. We were able to replicate the genes showing the signals under selection in the Mexican Indigenous populations, such as *IL17A*, *TFAP2B*, *NCKAP5L*, *ARHGAP15*, *KCNQ3*, and *PHLPP1* [9,11], although specific selection sweeps in each region were identified (Figure 2, Appendix A). The Center, South, and Southeast shared the highest number of genes under selection (Figure 2C, Appendix A), reflecting the fact that neighboring populations are more likely to be genetically related to each other than to distant groups [8,9,32].

Additionally, in all the regions, we found a strong enrichment of the selected genes involved in the metabolic and immune phenotypes (Appendix A). Although the set of genes under selection was different in each region, reflecting a potentially regional variability in the adaptive process, these data show a shared selection in the targets at the level of the biological pathways, which has been demonstrated in previous interpopulation comparisons [11,12,14,27].

Regarding the specific selection signatures in the genes related to the immune response, most of these genes were shared among the geographically close regions (Appendix A), but others were restricted to a single region: *IL17A*, *IL17F*, and *FcER1A* were found only in the North; *MIR146A* in the Northwest; *PRDM1* in the Center; *CD86* in the South; and *TNFSF8* in the Southeast. The presence of the natural selection signals in the immune-related genes can be explained in part by the demographic impact of the European colonization on Native American populations [7,11,33]. Previous studies have suggested that sustained epidemics of diseases, such as smallpox, measles, and “Huey Cocoliztli”, resulted in the strong selection for alleles that enhance the resistance to infections, changing the genomic landscape of Native Americans [6,7,34,35]. It is important to consider that the strong bottleneck caused by European colonization in the Americas could also affect the allele frequency of certain SNVs, including those located in genes related to the immune system [7,8,34]. Additional studies are necessary to confirm these hypotheses.

It has also been suggested that some genes involved in metabolic pathologies could be targets of natural selection in humans, because most of them affect the metabolism and energy production [36,37]. Some examples are SNVs at *CREBRF* and *FADS1* that are associated with obesity, type 2 diabetes, and lipemic traits that are under selection in Samoans and Europeans, respectively [25,38,39]. In this study, *PPARG* and *AJAP1* harbored selection signals common to all five regions. Notably, both genes have been related to the negative regulation of the Wnt/β catenin pathway, which is involved in the regulation of the metabolism, adipogenesis, and energy homeostasis and has been associated with the risk for obesity and type 2 diabetes [40]. In light of these observations, we hypothesize that the selection processes in *PPARG* and *AJAP1* could be the result of an adaptation to the reduced caloric intake during famines that Native American populations experienced through their history [11,12]. More studies are needed, both to elucidate the complex evolutionary history of the *AJAP1* and *PPARG* selection and to understand the population distributions of the relevant SNVs of these metabolic-related disease risk genes.

In addition to the terms related to the immune response and metabolic phenotypes, the enrichment analysis revealed other important phenotypes, such as the gut microbiome measurement, in all the regions (Appendix A). Although several studies have pointed out that the relative gut microbiome composition is mainly driven by demographic and environmental factors, it is now accepted that a host genetic variation can affect it to some degree [28,29,30,41]. A classic example of a locus under selection is the lactase gene (*LCT*), which also affects the relative abundance of *Bifidobacterium* (phylum *Actinobacteria*) in the European population [30]. Our data show that some genes previously suggested to influence the OTU abundance, prevalence, and alpha/beta diversity and the OTU prevalence of the human gut microbiome are under selection in all the studied populations across Mexico, displaying specific patterns in each region, possibly reflecting the differences in dietary regimes in different regions (Appendix A). Notably, we found region-specific genes under selection. For example, variants of *SLC6A11*, which affect the abundance of the genera *Oscillospira* and *Barnesiella* [28,29], were found in the Center, South, and Southeast, while variants of *TMTC2*, related to the abundance of the genus *Faecalibacterium* [30], were observed in the North and Northwest (Appendix A). These findings suggest that, together with demographic and environmental factors, the presence of these alleles under local adaptation in Indigenous populations from different regions could drive the local composition of the gut microbiome. This highlights the importance of studying how humans have coevolved with the microbial communities that colonize them in response to environmental factors.

The further elucidation of region-specific signals of selection, combined with the study of environmental factors that have influenced their genetic structure, could contribute not only to a better explanation of human evolutionary history but also to improved epidemiological and health policies.

## Figures and Tables

**Figure 1 genes-13-02251-f001:**
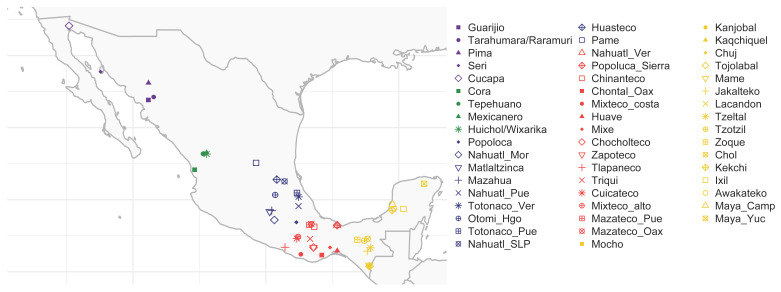
Geographical distribution of the 53 Mexican Indigenous communities included. Dot shapes denote the ethnic group and colors their assigned geographic region. Purple: North, green: Northwest, blue: Center, red: South, and yellow: Southeast.

**Figure 2 genes-13-02251-f002:**
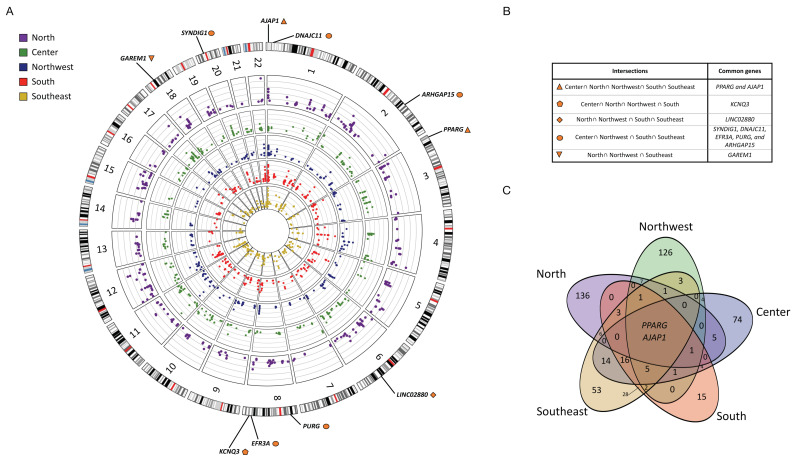
Significant adaptation signals in all tested regions. (**A**) SNVs under the top 1% distribution of iHS and PBS values in each region, only PBS values are plotted. (**B**) Genes symbols of selection signals shared by two or more regions. (**C**) Venn diagram showing both the number of or shared genes in all regions.

**Figure 3 genes-13-02251-f003:**
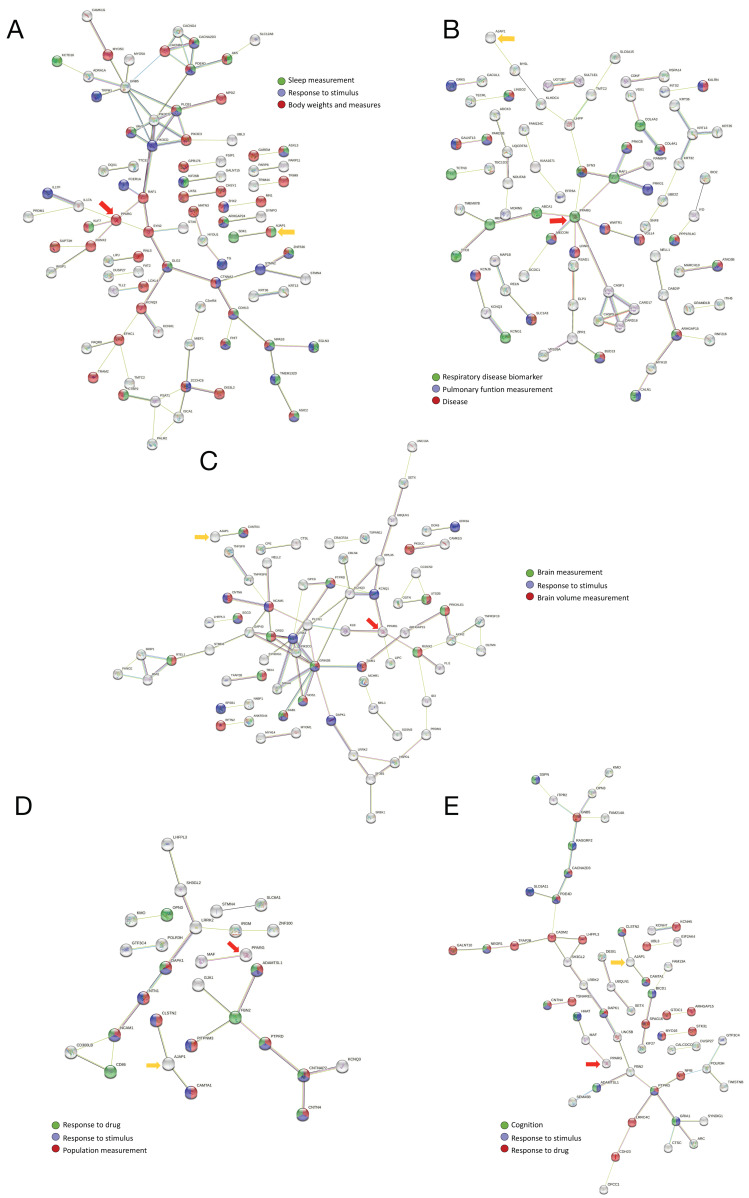
PPI networks constructed with significant adaptation signals observed in each region; PPARG and AJAP are highlighted with red and yellow arrows, respectively. (**A**) North. (**B**) Northwest. (**C**) Center. (**D**) South. (**E**) Southeast.

## Data Availability

We do not generate new genotype data in this project.

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
