# Peer review of "Unraveling Signatures of Local Adaptation among Indigenous Groups from Mexico"

_genes, 2022, doi:10.3390/genes13122251_

Round 1

Reviewer 1 Report

The article is of quality and interest. I only have a few minor modifications/suggestions:

Introduction, p 2, line 51, the authors say Recent studies but some of the cites [1-3] are from 1979. So, please revise either the sentence or the references.

Samples, p. 2, line 98. The sample from Peruvians in 1KGP phase 3 is composed of 33 individuals. Has any selection criteria been applied to select these 33 individuals? It sounds to me that there are more than 33 Peruvian individuals in the original database of 1KGP. Why Peruvian individuals instead of other American samples? I intuit the reason for the selection but it is better to make it clear in the text.

Discussion, as the authors state sustained epidemics of diseases resulted in strong selection for alleles that enhance resistance to infections (p. 6, lines 208-210). Unfortunately, these epidemics also resulted in sharp demographic declines in the population. Do the authors believe that these population reductions can affect their results? should it be taken into account in some way?

Author Response

Thank you very much for your helpful suggestions and your kindly comments on our manuscript. Below please find (in blue) a more detailed responses to each of your comments.

The article is of quality and interest. I only have a few minor modifications/suggestions:

Introduction, p 2, line 51, the authors say Recent studies but some of the cites [1-3] are from 1979. So, please revise either the sentence or the references.

Answer: You are right, there was a mistake. In the new version the references were updated.

Samples, p. 2, line 98. The sample from Peruvians in 1KGP phase 3 is composed of 33 individuals. Has any selection criteria been applied to select these 33 individuals? It sounds to me that there are more than 33 Peruvian individuals in the original database of 1KGP. Why Peruvian individuals instead of other American samples? I intuit the reason for the selection but it is better to make it clear in the text.

Answer: Those 33 Peruvians were selected based on their high Native American ancestry. To clarify the selection criteria of these 33 Peruvians in the main text in line 101 as follows:

“… the 33 Peruvians with the highest Native American Ancestry [12]”

Discussion, as the authors state sustained epidemics of diseases resulted in strong selection for alleles that enhance resistance to infections (p. 6, lines 208-210). Unfortunately, these epidemics also resulted in sharp demographic declines in the population. Do the authors believe that these population reductions can affect their results? should it be taken into account in some way?

Answer: Discussion, lines 256-258. In order to enrich the discussion about this topic, we added the following sentences:

“It is important to consider that the strong bottleneck caused by European colonization in the Americas could also affect the allele frequency of certain SNVs, including those located in genes related to the immune system [7,8,34].”

Reviewer 2 Report

The paper is clearly written. The authors have pointed out the most important information concerning concepts in the literature, the chosen individuals, their biological samples, as well as the methodology for the studies.  The use of individuals with at least 99% of Native American ancestry was approprieted for their purposes and objectives. The authors have done well considerations with respect to genetic changes, metabolic, immunological adaptations, gut selection microbiome, for instance, in response to environment pleasures as changes in the life style, long periods of starvation, european colonization and new diseases.

I guess the authors should clarify the meaning of the initials SNV (line 87).

Author Response

Thanks a lot for take the time to read and comment about our work. Below please find (in blue) a response for each of your comments.

The paper is clearly written. The authors have pointed out the most important information concerning concepts in the literature, the chosen individuals, their biological samples, as well as the methodology for the studies.  The use of individuals with at least 99% of Native American ancestry was approprieted for their purposes and objectives. The authors have done well considerations with respect to genetic changes, metabolic, immunological adaptations, gut selection microbiome, for instance, in response to environment pleasures as changes in the life style, long periods of starvation, european colonization and new diseases.

Thanks a lot for your comments about our work,  we are very grateful.

I guess the authors should clarify the meaning of the initials SNV (line 87).

Answer: The meaning of the initials SNV was added in line 89

Reviewer 3 Report

In this manuscript, Garcia-Ortiz and colleagues calculated integrated haplotype scores and population branch statistics in 325 Mexican Indigenous individuals from five geographic regions spanning the entire country in order to find novel signatures of local adaptation. They report signals in regions related to metabolic and immune processes, as well as gut microbiome adaptations. The novelty of the study lies in the examination of purely indigenous genomes as opposed to previous efforts that focused on admixed individuals.

The manuscript is very well-written, and the methods are adequate, concise, and nicely explained; the significance criteria are rather on the stricter side, which is good as it helps to avoid false discoveries and to make the findings much more credible. The results are presented in sufficient detail with nice Figures and the discussion includes the necessary element to help the reader interpret the findings.

My comments are below:

1.     lines 79-80: it would be nice to add - in one phrase - the method by which indigenous ancestry was ascertained previously. Was it by questionnaires? Or a genetic analysis like ADMIXTURE?

2.     What was the criterion for the division of the indigenous populations in the five geographic regions?

3.     Speaking of geographic regions, this paper is about indigenous populations in Mexico, yet the main text is stripped of any specific naming of the indigenous peoples studied. I think given how rigorous and well-writen the paper is, it is worth adding some more information about the actual populations studied in the main text. For instance, in lines 81 and 82, the authors can mention in parentheses the actual names of the regions studied and maybe the names of the Native groups living in them. Or, this is just a suggestion, even add a map in the main text to show the location of these populations.

4.     The authors interpret some of their findings in the context of European contacts. This means that, for this to be true, some of the adaptation signals found must be fairly recent. It would therefore be useful to provide some more context to the timeframe of these adaptations. For instance, iHS picks up extended haplotypes that are abundant, which by definition correspond to recent events. If the authors prefer to not run specific analyses for the dating of the adaptive signals and thus resolve this with hard numbers, they could discuss their findings in light of what the iHS method actually does.

5.     Even though there are previous analyses on admixed individuals, as correctly stated in the Introduction and Discussion, the authors do not make any comparisons/references between the two sets of findings. I think the paper could benefit from some more information on where the new results stand in comparison with previous findings, using maybe 1-2 specific examples. 

6.     The authors mention some specific microbiome taxa found to be enriched in their analysis. What do we know about these taxa? Can the authors add some information or speculation about how they would benefit their hosts?

Minor points:

1.     line 133: you probably mean Venn diagram? Please change accordingly.

2.     The resolution of Figure 2 is not big enough and the labels are not readable. Can the authors make sure there is big enough resolution for the printed version?

Author Response

Thanks a lot for your helpful suggestions and your kindly words on our manuscript. In response to these thoughtful suggestions, we have revised the manuscript to address all your comments. Below please find (in blue) a more detailed responses to each of your comments.

In this manuscript, Garcia-Ortiz and colleagues calculated integrated haplotype scores and population branch statistics in 325 Mexican Indigenous individuals from five geographic regions spanning the entire country in order to find novel signatures of local adaptation. They report signals in regions related to metabolic and immune processes, as well as gut microbiome adaptations. The novelty of the study lies in the examination of purely indigenous genomes as opposed to previous efforts that focused on admixed individuals.

The manuscript is very well-written, and the methods are adequate, concise, and nicely explained; the significance criteria are rather on the stricter side, which is good as it helps to avoid false discoveries and to make the findings much more credible. The results are presented in sufficient detail with nice Figures and the discussion includes the necessary element to help the reader interpret the findings.

My comments are below:

  1. lines 79-80: it would be nice to add - in one phrase - the method by which indigenous ancestry was ascertained previously. Was it by questionnaires? Or a genetic analysis like ADMIXTURE?

Answer: Ancestry proportion was previously inferred by ADMIXTURE (African, European and Native American). To be clearer, we also modified the Methods section at lines 80-82 as follows:

“Individuals having at least 99% Native American ancestry previously inferred by ADMIXTURE analysis, were selected from the MAIS Cohort [8]”.

  1. What was the criterion for the division of the indigenous populations in the five geographic regions?

Answer: According with a previous work from our group (García-Ortiz, et al., 2021, doi: 10.1038/s41467-021-26188-w.), a Neighbor Joining Tree analysis, based on pairwise FST values, showed that Indigenous groups from Mexico can be clustered in these five regions.

To be clearer, we added the following text to the Methods section at line 82-84:

“Furthermore, a Neighbor-Joining Tree based on pairwise fixation index (FST) values, showed that those Indigenous groups from Mexico can be clustered in five regions: North, Northwest, Center, South and Southeast [8].”

  1. Speaking of geographic regions, this paper is about indigenous populations in Mexico, yet the main text is stripped of any specific naming of the indigenous peoples studied. I think given how rigorous and well-writen the paper is, it is worth adding some more information about the actual populations studied in the main text. For instance, in lines 81 and 82, the authors can mention in parentheses the actual names of the regions studied and maybe the names of the Native groups living in them. Or, this is just a suggestion, even add a map in the main text to show the location of these populations.

Answer: Following your suggestion, we generated the new figure (Figure 1), showing a map including the location of all ethnic groups included in the study and their assigned geographic region.

  1. The authors interpret some of their findings in the context of European contacts. This means that, for this to be true, some of the adaptation signals found must be fairly recent. It would therefore be useful to provide some more context to the timeframe of these adaptations. For instance, iHS picks up extended haplotypes that are abundant, which by definition correspond to recent events. If the authors prefer to not run specific analyses for the dating of the adaptive signals and thus resolve this with hard numbers, they could discuss their findings in light of what the iHS method actually does.

Answer: Thank you very much for your suggestion. We think that although iHS detects recent selection sweeps, however, as far as we know iHS is no able to discriminate between recent and old selection sweeps.

We would like to mention that we are also preparing two papers on the dating of some of the identified haplotypes.

  1. Even though there are previous analyses on admixed individuals, as correctly stated in the Introduction and Discussion, the authors do not make any comparisons/references between the two sets of findings. I think the paper could benefit from some more information on where the new results stand in comparison with previous findings, using maybe 1-2 specific examples. 

Answer: Following your suggestion, in the Discussion section at lines 234-36 we added the next sentence:

“We were able to replicate genes showing signals under selection in Mexican Indigenous populations such like IL17A, TFAP2B, NCKAP5L, ARHGAP15, KCNQ3 and PHLPP1 [9, 11]”

  1. The authors mention some specific microbiome taxa found to be enriched in their analysis. What do we know about these taxa? Can the authors add some information or speculation about how they would benefit their hosts?

Answer: Thanks for the suggestion. To enrich our findings, in Table S18 we added a column with information reported in the literature for each taxa mentioned in our paper.

Minor points:

  1. line 133: you probably mean Venn diagram? Please change accordingly.

Answer: You are right, it was a typing mistake. The text was corrected appropriately.

  1. The resolution of Figure 2 is not big enough and the labels are not readable. Can the authors make sure there is big enough resolution for the printed version?

Answer: To improve the resolution of Figure 2 (now Figure 3) we increased the size as much it was possible. To avoid that labels were not readable in a printed version, we added all PPI networks as individual Supplementary figures (Figures S1-S5).

Reviewer 4 Report

The authors performed genome-wide scans using two common approaches (iHS and PBS) to identify genomic regions with positive selection in 5 regions of Mexico. They provide lists of genes detected for each of the 5 regions. They also investigate the pathways in which those genes are involved.

Despite the interest of identifying genes involved in local adaptation, there are parts of this manuscript that should be improved, mainly regarding the description of the the samples included, the methodology used as well as the interpretation of the results. There are some parts that need to be better explained, described or interpreted.

Comments:

-          In the introduction (lines 56-58) the authors cite some recent studies about selection, but they only mention those focused on Latin American populations, which are mainly admixed ones, but do not say anything else about the work referenced with number 10, which was focused on Native Americans. It would be good that the authors explain better which is the current state of the subject, what other studies did and what the present manuscript provides/adds in such context.

 -          The analysis was performed based on 5 geographic regions, but the authors do not mention why the analyses of positive selection considered these five regions and no another clustering. Which are the criteria to divide the populations into those 5 geographic regions? Is it because  of differentiated environmental conditions? How is the genetics and linguistic relationship of those groups? and of the populations within each group?.

I understand that the genotyping and population structure of those populations have been published (ref nº 8), but the main findings of that work should be included in the present manuscript for the reader to understand why the analyses were performed using those groups.  For example, how homogeneous the populations included in each region are. Are there differences between the different regions?

-          In this sense, it would be appreciated to have a map with the location of the populations and the division of the regions, and a paragraph indicating the most important aspects of those populations, genetically, linguistically, and environmentally.

- Is there an hypothesis of what would be expected?

-  This issue is also related with the discussion section, where there is no connection of results obtained for each region and the environmental, cultural, historical particularitites of those regions that could have led to obtain such results. Thus, I would suggest that in addition to just provide list of genes, the text should include more details about how to interpret the results that have been obtained.

- Regarding the methodology, more details should be included. How many SNPs were considered?

- The genes that appear as having signs of positive selection, how were they identified? Are the selected SNPs within those gene regions? or the genes are the closer ones to the SNPs? Is there a particular distance between the SNP and a gene to be considered as the putatively targeted one? There is no information about important aspects of the methodology that should be included.

- Line 166: The authors mention: “One of the overrepresented biological pathways observed in all regions was the gut microbiome measurements (Table S8‒S12)”. However, there are other overrepresented biological pathways with even more significance that are not commented. Why do the authors put so much emphasis on the gut microbiome? If they think that this aspect is that important, it should be related with the particular characteristics of the populations studied.

- Line 173.  Related to this, the authors mention that the genetic profile of each region that could influence a different gut microbiome ("These data reflect possible differences in gut microbiome composition according to geographic location and also the genetic profile of each region"). But in the manuscript there is no information about which is the genetic relationship between the populations and regions included, nor whether there are cultural or environmental differences between those regions that could influence possible differences. 

- This lack of information, and a better interpretation of the results is also related with the fact that the title mentions Local adaptation, and the analysis was performed based on 5 geographical regions, then, I would expect that there is some more explanation of the reason why to consider those 5 regions separately, as well as on the results obtained for each particular region, that is not just a list of genes in a supplementary table. 

Minor comments:

-          Line 56: The authors use the term “great” diversity observed in Native American populations [4–8]. However, considering that Native Americans are the populations showing the les genetic diversity compared to other world populations, the term “great” seems to me, inadequate. I suggest removing it.

-          Line 63: The authors mention “certain diseases”, I think that at least one or two should be named. 

Author Response

Thanks a lot for take the time to read and comment about our work. Below please find (in blue) a more detailed responses to each of your comments.

The authors performed genome-wide scans using two common approaches (iHS and PBS) to identify genomic regions with positive selection in 5 regions of Mexico. They provide lists of genes detected for each of the 5 regions. They also investigate the pathways in which those genes are involved.

Despite the interest of identifying genes involved in local adaptation, there are parts of this manuscript that should be improved, mainly regarding the description of the the samples included, the methodology used as well as the interpretation of the results. There are some parts that need to be better explained, described or interpreted.

Comments:

- In the introduction (lines 56-58) the authors cite some recent studies about selection, but they only mention those focused on Latin American populations, which are mainly admixed ones, but do not say anything else about the work referenced with number 10, which was focused on Native Americans. It would be good that the authors explain better which is the current state of the subject, what other studies did and what the present manuscript provides/adds in such context.

Answer: The paper by Amorim et al. 2017 includes Native Americans highly admixed previously published by Reich and Skoglund, as the same authors reports. Based on it, this publication was included in the group of papers that perform natural selection scan with admixed individuals. As far as we know, our paper is the first to publish this kind of studies in present-day individuals with 99% Native American ancestry.

-  The analysis was performed based on 5 geographic regions, but the authors do not mention why the analyses of positive selection considered these five regions and no another clustering. Which are the criteria to divide the populations into those 5 geographic regions? Is it because of differentiated environmental conditions? How is the genetics and linguistic relationship of those groups? and of the populations within each group?.

Answer: According with a previous work from our group (García-Ortiz, et al., 2021, doi: 10.1038/s41467-021-26188-w.), a Neighbor Joining Tree analysis, based on pairwise FST values, showed that Indigenous groups from Mexico can be clustered in these five regions.

To be clearer, we added the following text to the Methods section at line 82-84:

“Furthermore, a Neighbor-Joining Tree based on pairwise fixation index (FST) values, showed that those Indigenous groups from Mexico can be clustered in five regions: North, Northwest, Center, South and Southeast [8].”

On the other hand, García-Ortiz, et.al., 2021 also includes a linguistic and genetic structure comparison. We believe that it is not necessary to add it, since it is discussed in our previous paper and referenced throughout the present manuscript (Ref. 8).

I understand that the genotyping and population structure of those populations have been published (ref nº 8), but the main findings of that work should be included in the present manuscript for the reader to understand why the analyses were performed using those groups.  For example, how homogeneous the populations included in each region are. Are there differences between the different regions?

- In this sense, it would be appreciated to have a map with the location of the populations and the division of the regions, and a paragraph indicating the most important aspects of those populations, genetically, linguistically, and environmentally.

Answer: Following your suggestion, we generated the new figure (Figure 1), showing a map including the location of all ethnic groups included in the study and their assigned geographic region. These with the included the text at lines 82-84, mentioned in the response to your previous commentary brings a better description of the population’s classification.

- Is there an hypothesis of what would be expected?

Answer: There is not a hypothesis because it is a descriptive analysis, but we aimed to understand how selective pressure has shaped the genetic structure of Native American populations without the potential bias of recent admixture.

-  This issue is also related with the discussion section, where there is no connection of results obtained for each region and the environmental, cultural, historical particularitites of those regions that could have led to obtain such results. Thus, I would suggest that in addition to just provide list of genes, the text should include more details about how to interpret the results that have been obtained.

Answer: We agree with you; this issue was solved with the inclusion of Figure 1. On the other hand, we believe that those issues are discussed at lines 251-59 as follows:

“The presence of natural selection signals in immune-related genes can be explained in part by the demographic impact of European colonization on Native American populations [7,11,33]. Previous studies have suggested that sustained epidemics of diseases such as smallpox, measles, and “Huey Cocoliztli” resulted in strong selection for alleles that enhance resistance to infections, changing the genomic landscape of Native Americans [6,7,34,35]. In addition, the strong bottleneck caused by European colonization in the Americas could also affect the allele frequency of certain SNVs, including those located in genes related to the immune system [7,8,34]. Additional studies are necessary to confirm these hypotheses.”

And in lines 260-73:

“In this study, PPARG and AJAP1 harbored selection signals common to all five regions. Notably, both genes have been related to negative regulation of the Wnt/beta catenin pathway, which is involved in the regulation of metabolism, adipogenesis, and energy homeostasis and has been associated with risk for obesity and type 2 diabetes [40]. In light of these observations, we hypothesize that selection processes in PPARG and AJAP1 could be the result of adaptation to reduced caloric intake during famines that Native American populations experienced through their history [11,12]. More studies are needed, both to elucidate the complex evolutionary history of AJAP1 and PPARG selection and to understand the population distributions of relevant SNVs of these metabolic-related disease risk genes.”

- Regarding the methodology, more details should be included. How many SNPs were considered?

Answer: Thanks for your suggestion, in methods section at line 86 we added the following sentence:

“A total of 654,023 SNVs were included for genetic analyses.”

- The genes that appear as having signs of positive selection, how were they identified? Are the selected SNPs within those gene regions? or the genes are the closer ones to the SNPs? Is there a particular distance between the SNP and a gene to be considered as the putatively targeted one? There is no information about important aspects of the methodology that should be included.

Answer: In the section 2.3 of the methods we stated:

“Variants considered under selection in each region were annotated by the Ensembl Variant Effect Predictor [21].”

That Predictor includes all issues you mention, as stated in the VEP main page: “to determines the effect of your variants (SNPs, insertions, deletions, CNVs or structural variants) on genes, transcripts, and protein sequence, as well as regulatory regions. Simply input the coordinates of your variants and the nucleotide changes to find out the: Genes and Transcripts affected by the variants Location of the variants (e.g. upstream of a transcript, in coding sequence, in non-coding RNA, in regulatory regions) Consequence of your variants on the protein sequence (e.g. stop gained, missense, stop lost, frameshift), see variant consequences Known variants that match yours, and associated minor allele frequencies from the 1000 Genomes Project SIFT and Polyphen score for protein sequences” https://www.ensembl.org/info/docs/tools/vep/index.html

-Line 166: The authors mention: “One of the overrepresented biological pathways observed in all regions was the gut microbiome measurements (Table S8‒S12)”. However, there are other overrepresented biological pathways with even more significance that are not commented. Why do the authors put so much emphasis on the gut microbiome? If they think that this aspect is that important, it should be related with the particular characteristics of the populations studied.

Answer: You are right, we put more emphasis on the gut microbiome because of the novelty of our result in this emergent field, and as far as we know, it is one of the first reports that shown a direct correlation between genes under selection and gut microbiome composition.

However, other overrepresented biological pathways have been also discussed in other studies that we are referring (24-27).  We mentioned them in the Results section of our paper as follows:

“Among them, the immune response, metabolic pathways, cardiovascular phenotypes, central nervous system, chemical dependence, response to stimulus, pulmonary function, hematological conditions, body weight, brain and gut microbiome measurements, were the common terms found to be enriched in the putatively selected genes in all regions (Table S8‒S17). However, the genes enriching the pathways were different from those reported for other populations, such as European or Japanese populations [24–27]. Some are exclusive to one or more geographic region, revealing putative signatures of local adaptation in Mexican Indigenous populations (Figure 2C; Table S8‒S17).”

- Line 173.  Related to this, the authors mention that the genetic profile of each region that could influence a different gut microbiome ("These data reflect possible differences in gut microbiome composition according to geographic location and also the genetic profile of each region"). But in the manuscript there is no information about which is the genetic relationship between the populations and regions included, nor whether there are cultural or environmental differences between those regions that could influence possible differences.

Answer: You are right, we believe that these topics are discussed in lines 274-93 as follows:

“In addition to terms related to immune response and metabolic phenotypes, the enrichment analysis revealed other important phenotypes, such as gut microbiome measurement, in all regions (Table S8-S12 and Table S18). Although several studies have pointed out that relative gut microbiome composition is mainly driven by demographic and environmental factors, it is now accepted that host genetic variation can affect it in some degree [28–30,41]. A classic example of a locus under selection is the lactase gene (LCT), which also affects the relative abundance of Bifidobacterium (phylum Actinobacteria) in the European population [30]. Our data show that some genes previously suggested to influence OTU abundance, prevalence, and alpha/beta diversity, and OTU prevalence of the human gut microbiome are under selection in all studied populations across Mexico, displaying specific patterns in each region (Table S18). Notably, we found region-specific genes under selection. For example, variants of SLC6A11 which affect the abundance of the genera Oscillospira and Barnesiella [28,29], was found in the Center, South, and Southeast, while variants of TMTC2, related to the abundance of genus Faecalibacterium [30], was observed in the North and Northwest (Table S18). These findings suggest that, together with demographic and environmental factors, the presence of these alleles under local adaptation in Indigenous populations from different regions could drive the local composition of the gut microbiome. This highlights the importance of studying how humans have coevolved with the microbial communities that colonize them in response to environmental factors.”

Based on these results, we now have a new research project focused on the microbiome composition according to geographic location and the genetic profile.

- This lack of information, and a better interpretation of the results is also related with the fact that the title mentions Local adaptation, and the analysis was performed based on 5 geographical regions, then, I would expect that there is some more explanation of the reason why to consider those 5 regions separately, as well as on the results obtained for each particular region, that is not just a list of genes in a supplementary table.

Several changes to our manuscript based on your suggestions. We hope these changes addressed all your concerns and found the manuscript of enough quality.

Minor comments:

-          Line 56: The authors use the term “great” diversity observed in Native American populations [4–8]. However, considering that Native Americans are the populations showing the les genetic diversity compared to other world populations, the term “great” seems to me, inadequate. I suggest removing it.

Answer: Thank you for your suggestion, we removed the term great from the main text.

-          Line 63: The authors mention “certain diseases”, I think that at least one or two should be named.

Answer: We added two examples in the Introduction section at line 63 as follows:

“such like metabolic and immune entities.”

Round 2

Reviewer 3 Report

The authors have addressed all the issues I raised adequately. I have no further comments.

Reviewer 4 Report

The authors have addressed some of the comments and the manuscript has improved. However, there is still a lack of information regarding differences in environment, diet, or any factor that could explain or at least give a clue on possible reasons for differences between the five regions. To me this is important, even if it is not explained in detail, because if there is no different environment, climate, diet between the regions, then all these differences could just be due to the effect of demography rather that local adaptation as it is mentioned in the title.

I would suggest to include a paragraph to mention possible differences, (climate, biomes, diet) that differenciate those regions and could be the reasons for this different adaptive adaptation. Otherwise, if there is no differences, only the genetic ones, I would removed "local adaptation" from the title: Unraveling Signatures of Positive Selection Among Indigenous Groups from Mexico.